# Sampling Techniques Affect Mayfly Nymph Community Indices and May Bias Bioassessments

**DOI:** 10.3390/insects16070723

**Published:** 2025-07-16

**Authors:** Zohar Yanai, Netta Dorchin

**Affiliations:** 1The Steinhardt Museum of Natural History, Tel Aviv University, Tel Aviv 6997801, Israel; 2School of Zoology, Faculty of Life Sciences, Tel Aviv University, Tel Aviv 6997801, Israel; ndorchin@tauex.tau.ac.il

**Keywords:** biomonitoring, Ephemeroptera, field survey, hand net, methodology, stream

## Abstract

Freshwater habitats are monitored and assessed for their integrity, with their associated invertebrate community being a reliable indicator for water quality and habitat health. Invertebrate collection techniques may affect the resulting assemblage—and hence the assessment conclusion. We compared two sampling techniques—net sweeping and stone collecting—for their performance, focusing on mayfly nymphs as model taxa, as they are well known to be sensitive to diverse ecological factors. Our two techniques yielded inconsistent results for various biological indices, suggesting that assessment conclusions may be subject to bias. In addition, each technique was superior to the other technique in sampling certain mayfly species. We present some important considerations for choosing the best sampling technique for a given research objective based on its performance.

## 1. Introduction

Assessing environmental conditions (i.e., bioassessment) of aquatic systems, such as streams, is widely practiced by characterising the community structure of species-rich groups such as diatoms, fishes, and macroinvertebrates. The community composition and resulting ecological indices provide important information on a system’s stability and health, whilst species traits (e.g., tolerance or sensitivity to disturbance) contribute additional information [1,2]. The power of bioassessment stems from its comparability: within streams of a similar type, “healthy” assemblages serve as a reference for evaluating disturbances and setting restoration goals, and comparisons along temporal, spatial, or ecological gradients may reveal trends and processes [2,3,4,5]. When comparing communities, a key assumption is that they were sampled using similar appropriate methods. However, numerous sampling protocols were designed and are currently used in various ongoing surveys in a way that may lead to different results and conclusions due to inherent differences [1,2,6]. Sampling methods may have a fundamental effect on sampled assemblages [7,8,9] and consequently on the conclusions drawn from them. Therefore, it is important to recognise the constraints of each sampling method and select the most suitable one or at least be aware of its implications for the results.

Macroinvertebrates, which are arguably the most widely studied and commonly used taxa for bioassessments of freshwater habitats, can be sampled using a variety of methods, many of which were compared in previous studies (e.g., [6,7,8,10]). Mayflies (order Ephemeroptera) usually play a central role in aquatic biological indices, because their aquatic nymphs are relatively easily collected and identified and because they exhibit a variety of functional traits. At the same time, most mayflies are highly sensitive to various disturbance types, such as pollution and structural alterations in the habitat [11,12]. These attributes render them reliable indicators of ecological changes, either as the main components in community-based indices or as focal taxa for study. Field ecologists can learn a lot from the presence, absence, and abundances of mayfly nymphs in streams, as long as this information is obtained in a standard manner and is comparable to other surveys.

In the present study we examined two sampling techniques. The first is net sweeping, by which an aquatic sampling net is used to sample macroinvertebrates across diverse microhabitats (e.g., substrate, vegetation). The catch is usually briefly screened in the field, transferred into ethanol, and subsequently sorted, identified, and counted in the lab. This is a popular sampling technique in many freshwater studies [1,7] and bioassessment protocols (e.g., [3,13,14,15]). The use of a D-shaped hand net is probably the most widely employed sampling method for benthic macroinvertebrates [1,16], often by sweeping, kicking, or a combination of both. The efficiency of this technique was tested in the past against other sampling methods, e.g., Surber sampling [6,10,17], Hess sampling [6,18], litterbags, corers [19], rock-bag sampling [8], Cretan shovels [9], and even against a traditional Māori fishing method [20].

The second method we tested is direct manual collection of individual specimens from submerged stones. This technique involves randomly picking stones of various sizes from the streambed and screening the biota that are attached to them. If quick enough, the sampler can use fine forceps to manually collect macroinvertebrates into a tube before they drop off the stone. This targeted sampling method results in clean specimens that can be inspected in the lab without further sorting. Like net collection, stone collection is very common and widely used, but usually as a supplement to other sampling protocols [13,16]. It is very seldom mentioned in protocols or used in standardised surveys, such as in a qualitative, non-standardised manner by Kerans et al. [6]. Despite its absence from widely used protocols, every stream ecologist or aquatic entomologist is well familiar with this method, as it is used haphazardly on virtually every visit to the field, regardless of the “formal” biomonitoring effort.

In order to assess the potential effect of sampling methods on the resulting assemblages, we conducted a field survey in which we employed the two sampling techniques and compared the conclusions derived from them, focusing on mayfly nymphs as a case study. We hypothesised that a sampling net will yield larger numbers of individuals (higher abundance), as well as higher taxonomic diversity. We expected to find differences in assemblages between the techniques, with clinger species that usually attach to the substrate being more common in stone samples whilst pelagic, free-swimming species and soft substrate dwellers would be more common in net samples. The handling time in the field was expected to be longer for the stone technique and much longer in the lab for the net technique.

## 2. Materials and Methods

### 2.1. Field Survey

The field survey was conducted in a setup of 15 perennial streams across Israel (one in the West Bank) that were visited as part of an ongoing ecological study (Yanai and Dorchin, in preparation): the Upper Jordan River and the streams Dan, Snir, Divsha, Tina, Jilabun, Kziv, Tsippori, Taninim, Alexander, Sorek, Qelt, David, Arugot, and Bokek. These streams were selected in order to represent the climatic and ecological diversity in the country, from Mediterranean to desert streams and from mountainous to lowland streams, with a wide range of human disturbance levels (e.g., pollution, damming). Prior experience showed that these streams accommodate different mayfly communities [21,22,23,24]. Since we sought to characterise the inter- and intra-stream variation and seasonal variability, each stream was sampled once in fall and once in spring, at two different sites. In most sites, a diversity of microhabitats was available, and we considered all of them in a proportional division to form one integrative sampling site. Each visit is referred to here as a sampling event, which included both net sampling and stone sampling, applied in a random order—i.e., half of the net effort, then half of the stone effort, and so on. Sampling was always conducted from the lowest point in the site, moving upstream, thus avoiding potential interference of the biota due to previous activity.

### 2.2. Net Sampling

Net collection (Figure 1a) was conducted using a D-frame water-collecting net for aquatic macroinvertebrates (ring size 30.5 × 25.5 cm, mesh size 500 µm; BioQuip, Gardena, CA, USA). Sampling was conducted along the banks and within the watercourse in an attempt to cover all available benthic and pelagic microhabitats (e.g., stony substrate, silt pockets, submerged vegetation, water column), aiming to represent the microhabitat composition in the site as accurately as possible. The net was used in one of two manners: either sweeping within the water column or against submerged objects, or kicking the stream bed into the stationary net when placed against the current for a total of 120 sweeps/kicks. We opted for a standardised number of sweeps as a more reliable measure than a standardised sampling time [13,17,19] or number of “replicates” [3]. Net sampling was performed by sweeping or kicking only, thus avoiding scrubbing stones directly into the net, as this would be too similar to the stone technique (see below). The entire net content was transferred to 70% ethanol on site.

### 2.3. Stone Sampling

Manual collection from stones and pebbles (Figure 1b) was carried out by picking them haphazardly from a variety of depths, current velocities, and other environmental features to reflect the composition of features on site. Ideally, screened stones should be similar in size and shape to allow for comparability, but complete similarity cannot be achieved in a natural setup without losing the attached organisms. Instead, we screened a fixed number of 60 fist-size stones at each site and manually collected all the mayflies that were found on them directly into a 50 mL tube containing 70% ethanol using fine forceps.

### 2.4. Lab Analysis

The collected material was sorted, identified, and counted in the lab. Mayfly identification was based on published checklists and identification keys for the local fauna (mainly [21,22,23]). Samples were identified to the species level, except for *Caenis* spp., for which large numbers of young nymphs were found and could not be identified because the four local species are sympatric and difficult to distinguish morphologically [24]. Two species from the *Cloeon dipterum* species group are undescribed, and a local *Procloeon* is not yet identified to the species level ([23]; J.-L. Gattolliat, Z. Yanai unpublished data). All specimens were deposited in the entomological collection of the Steinhardt Museum of Natural History (SMNH) at Tel Aviv University.

### 2.5. Statistical Analyses

To estimate the true number of mayfly taxa at each sampling event and ensure that our sampling effort was sufficient in detecting most of the present mayfly taxa, we used the non-parametric diversity estimator Jackknife1. We estimated the expected number of taxa for each sampling method separately in order to detect any potential gap in their ability to reflect the “true” taxon richness (regardless of species identity, which was addressed separately as detailed below).

We obtained information on biological and ecological traits from www.freshwaterecology.info/ (accessed on 31 March 2025) [25], including microhabitat preference (sensu [12]) and locomotion type (sensu [3,26]). We also estimated the saprobic index of the assemblages, given that saprobic indices reflect the species tolerance to organic load (pollution), which makes them an important tool in stream assessment schemes. The saprobic index is usually country-specific, and the Israeli version (ranking species on a scale of 1–10) is currently under construction by the Israel Center for Aquatic Ecology (Y. Hershkovitz, unpublished data). All the traits were set based on personal knowledge and on published data regarding related taxa (e.g., [25]). These traits were selected, as they were informative and useful in other bioassessments, and because they were possible to score for the Israeli mayfly fauna. Data on the local species or related species are scarce in the literature and therefore preclude analyses based on other traits.

Our final dataset included 53 paired sampling events composed of the net vs. stone techniques at the same site and time. Data were analysed only for sampling events that included at least two taxa regardless of sampling method, resulting in 44 relevant events. For the community structure analysis, two additional sampling events were removed, as they yielded no findings in the stone technique (no community to compare).

Diversity indices that are commonly used in bioassessments were calculated for all samples, including total abundance, taxon richness, Shannon–Wiener’s diversity index, and saprobic index. All scores were compared in a paired design, such that the net score and the stones score were compared for each assemblage. We used Wilcoxon’s test, since the distribution of differences between pairs was usually not normal. The total abundance was expected to differ considerably between methods but was analysed nonetheless, because both assemblages are in fact subsets of the total community; hence, they are expected to reflect the true abundance and proportions amongst taxa. To identify the most efficient sampling technique for a given taxon, the occurrence was also compared for each taxon separately.

The community structure, based on the log-transformed total abundance, was visualised with NMDS ordination. We calculated the Bray–Curtis dissimilarity amongst all samples separately for each method to yield two distance matrices. We then used the Mantel test to test for correlation between the two independent distance matrices. All statistical analyses were conducted in R v.4.1.1 [27] with the vegan package v.2.5–7 [28].

## 3. Results

Out of 53 sampling events, 9 yielded 0–1 taxa (net + stones) and were excluded from all further analyses. The remaining 44 paired sampling events were analysed as described below.

A total of 16,549 individuals were collected (range per sampling event: 8–2525 individuals; sd = 555.48) belonging to 20 mayfly taxa (range per sampling event: 2–8 taxa; sd = 1.34). The true number of taxa was estimated using the non-parametric diversity estimator Jackknife1 to be 20 taxa for all the samples combined, suggesting that all species that occurred in the study sites were indeed sampled. Each of the sampling methods reached the same cross-site taxon richness by itself, with estimates of 20.9 and 19.9 taxa for the net and stone techniques, respectively. These findings were also supported by accumulation curves (Figure 2).

The number of individuals obtained by both sampling techniques correlated well (Pearson’s correlation, r = 0.71; n_1,2_ = 44; *p* < 0.001; Figure 3a), suggesting that both were consistent in sampling a subset of the total community, although the non-selective net method yielded significantly more individuals (Paired *t*-test, t = 3.26; n_1,2_ = 44; *p* < 0.01; Figure 3b). This method also yielded a higher number of taxa (Wilcoxon signed-ranks test (paired design), V = 494; n_1,2_ = 44; *p* < 0.001; Figure 3c) and, as expected, the abundance of collected mayflies explained the number of taxa that were represented in each sampling event, such that the more individuals that we collected, the higher the taxon richness was expected to be (Spearman’s rank correlation, r = 0.51; n = 88; *p* < 0.001; Figure 3d). The Shannon–Wiener’s diversity index (*H*’), which was also calculated for all samples and compared between the two methods, indicated a significant difference between them (Wilcoxon signed-ranks test (paired design), V = 640; n_1,2_ = 44; *p* = 0.04; Figure 3e). The saprobic index for stone samples was usually higher than that for net samples (Wilcoxon signed-ranks test (paired design), V = 129; n_1,2_ = 42; *p* = 0.001; Figure 3f).

We compared the community structure in order to identify technique-related patterns. The community structure showed a similar pattern between techniques in some sites but was inconsistent in others (Figure 4a). The Bray–Curtis dissimilarity amongst all site pairs was generally greater for net samples compared to stone samples (Wilcoxon signed-ranks test (unpaired design), W = 424,045; n_1,2_ = 861; *p* < 0.001), suggesting that the net technique may be more species-sensitive. However, when comparing the pairwise dissimilarity within each distance matrix, they did not differ significantly (Mantel test with 9999 permutations, r = 0.69; *p* < 0.001; Figure 4b).

All taxa were collected using both methods with a few exceptions: *Cloeon* sp.1, *Procloeon* sp., and *Alainites gasithi* were not collected from stones, whereas *Ecdyonurus asiaeminoris* was not collected by net. These exceptions include the two rarest species; of the 20 species collected in this study, *E. asiaeminoris* was the only singleton, and *Procloeon* sp., *Choroterpes picteti*, and *Oligoneuriopsis orontensis* were doubletons. The most common taxa were *Caenis* sp. and *Baetis monnerati* (each found in 35 sampling events). Analysing the relative abundance of each taxon by sampling method revealed a significant association of three species to net collection (Table 1). For six additional species, the sample sizes were too small to run reliable statistical analyses but showed clear trends that enabled association with either of the sampling methods. The 11 remaining species were not significantly associated with either of the sampling methods. This mixed pattern suggests that specific attributes may affect the association of species with certain collection techniques.

## 4. Discussion

In the present study, we compared two commonly used sampling methods for mayfly nymphs, one of which is well documented in the methodological literature, whereas the efficiency and relevance of the other has rarely been tested. The two methods differ fundamentally by nature; whilst the comprehensive and inclusive net technique shifts most of the effort to the lab stage, the stone technique demands greater attention in the field in order to achieve only the most desired results (“cherry picking”). Both methods have merits and flaws (Table 2) and should be employed according to the research question. Net sweeping results in a large number of macroinvertebrates (Figure 3b) that include virtually all individuals captured during sampling. If carried out properly, this method is expected to yield a representative selection of the local biodiversity. The nymphs of many mayfly species are morphologically similar and can only be distinguished upon careful examination in the lab; others may hide within sediment and leaf litter and will only be revealed in the lab. Therefore, it is highly probable that additional species that were not observed in the field will raise the sample’s richness score following lab sorting. Indeed, sweep net sampling usually achieves the highest taxon richness scores compared to various other sampling techniques [7,19] (but see [9,10,17]). In contrast, when using the stones technique, one may focus on a taxon of interest but neglect to identify (and collect) other taxa that deserve closer attention in the lab (e.g., rare species, invasive species, or first locality records of previously undetected species). On the other hand, this method avoids the unnecessary collection of many macroinvertebrates in case of a focused study. In the present study, the risk of missing individuals of interest by stone sampling was minimised by the quick collection of all mayfly nymphs that were found on stones, yet the taxon richness was still significantly lower compared to net sweeping.

For ecological studies that aim to evaluate entire communities, the net technique is preferable to the stone technique, as it is likely to yield a more realistic taxon richness score. However, for comparison amongst samples and the identification of spatial and temporal patterns, the two methods are equally useful. Despite the lower richness in stone samples, the overall community structure resembled that derived from net sampling. Shannon–Wiener’s diversity index (*H*’) showed the same pattern, with the higher taxon richness probably resulting in higher *H*’ values in net samples. This measure naturally depends on the taxon richness and may lead to inconsistent results (see [8]), suggesting that none of the methods artificially alter *H*’. Moreover, in order to obtain the most representative results, it is crucial to employ a method (or combine several methods) that can be used in various habitat types (e.g., riffles and pools), which may accommodate different species [6,29]. In the present study, neither method yielded all the available species per sampling event, emphasising the power of combining them.

Some species may be collected much more often with one technique (Table 1). Whilst this finding highlights the importance of employing the correct method for a given research question, in our study, nearly all taxa were retrieved by both techniques. In contrast, Muzaffar and Colbo [8] found vast differences between the two collection methods that they tested, perhaps because they sampled a larger variety of macroinvertebrates. Indeed, when considering mayflies alone, the sampling technique did not have much effect in that study, with the exception of *Caenis simulans*, which was effectively collected using net sweeping, similarly to *Caenis* spp. in our study (Table 1).

The saprobic index provides vital information for ecological surveys, as it reflects the ability of a community to tolerate organic pollution [30]. *Cloeon* and *Caenis* species, which are associated with lower saprobic values, dominated our net samples, whereas heptageniids (typical of higher saprobic values) were found more often in stone samples. Since many of the commonly used bioassessment protocols employ only the net technique, the resulting evaluations may be biassed towards poorer water quality.

Net sweeps performed better for species that are found in soft sediments or swim freely (e.g., *Cloeon*, *Procloeon*, and *Caenis* spp.; Table 1). These are often disturbed or drawn out of their hide by the sweep and then easily collected [8]. The stone technique, however, is naturally better for spotting and collecting rock-dwellers such as *Ecdyonurus* and *Epeorus*, which are sprawlers that are often found in coarse gravel and cobbles (Table 1). This pattern corresponds to the findings of similar methodological studies, in which the stone-associated Surber sampling was used [10,17]. A surprising deviation from this pattern was found in the case of *Rhithrogena znojkoi* (Heptageniidae), a typical rock-dweller, which was more abundant in net samples. This finding led us to take a closer look at the collected material, and to the finding that net-collected individuals were much younger and smaller than stone-collected conspecifics. This is probably because early instar nymphs of this species are weaker clingers than mature ones (ZY, self-observation) and therefore easier to remove from rocks into a sweeping net, or because they drop off a stone when picked. This issue should be taken into account if one wishes to collect a rock-dwelling taxon for a focused study, as it probably means that only mature, larger specimens will be sampled. Nevertheless, if one seeks to compile a complete taxon checklist for ecological evaluations, the high numbers of early instar nymphs may be more significant.

Rare taxa collected here in the fewest sites and lowest numbers may be missed if the wrong sampling method is employed, although this will probably have a limited impact on community-level results [31]. However, many sampling schemes seek to track the rare, often sensitive or endangered, species as indicators of the impact of disturbances or the success of restoration efforts [17]. Given that each method was more sensitive to different rare taxa (Table 1), it is highly recommended to use both (and perhaps, additional) methods for the purpose of ecological surveys.

An additional point of practical importance that we did not quantify or analyse but is worth mentioning is the fact that the net technique is much more aggressive than the targeted stone technique and entails the transfer of samples to the laboratory in a large volume of ethanol. These conditions often damage fragile invertebrates, for example by breaking body parts of taxonomic value (e.g., legs and gills). In contrast, stone sampling usually demands careful attention in the field and is therefore more time-consuming, but the resulting samples are of a higher taxonomic quality. Time and budget are often limited resources in scientific research and ecological surveys; hence, these considerations should also be factors in a study design.

## 5. Conclusions

Based on our findings, we propose points for consideration for selection of the most proper method for mayfly nymph (and other macroinvertebrate) collection (Table 2). Whilst our results are local and suit a limited pool of taxa, these principles can easily be adjusted to fit surveys of other geographical and taxonomic foci. We recommend considering the combination of these two commonly practiced methods when developing future protocols, given that the current protocols [3,13,14] may miss important biota. Despite the common use of stone sampling [16], only a few protocols (e.g., [15]) recommended its use, and even then only for a limited time or for collection of focal taxa in order to complement net sampling. We recommend to always incorporate a survey of a standardised number of stones, from which all observed individuals should be collected, in order to produce a quantitative, measurable, and comparable dataset. Naturally, when sampling a site for the first time or following a significant environmental alteration, it is best to employ both methods in order to record as many biological features as possible.

## Figures and Tables

**Figure 1 insects-16-00723-f001:**
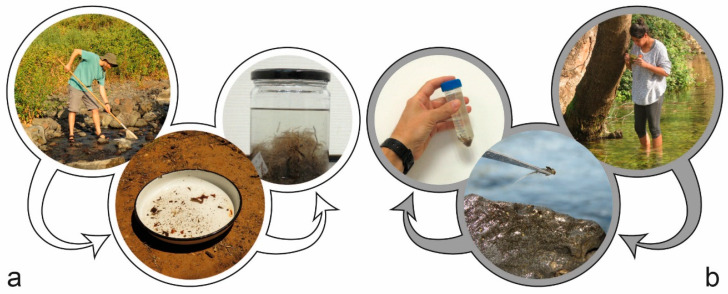
Sampling methods. (**a**) Hand net; (**b**) manual collection from stones.

**Figure 2 insects-16-00723-f002:**
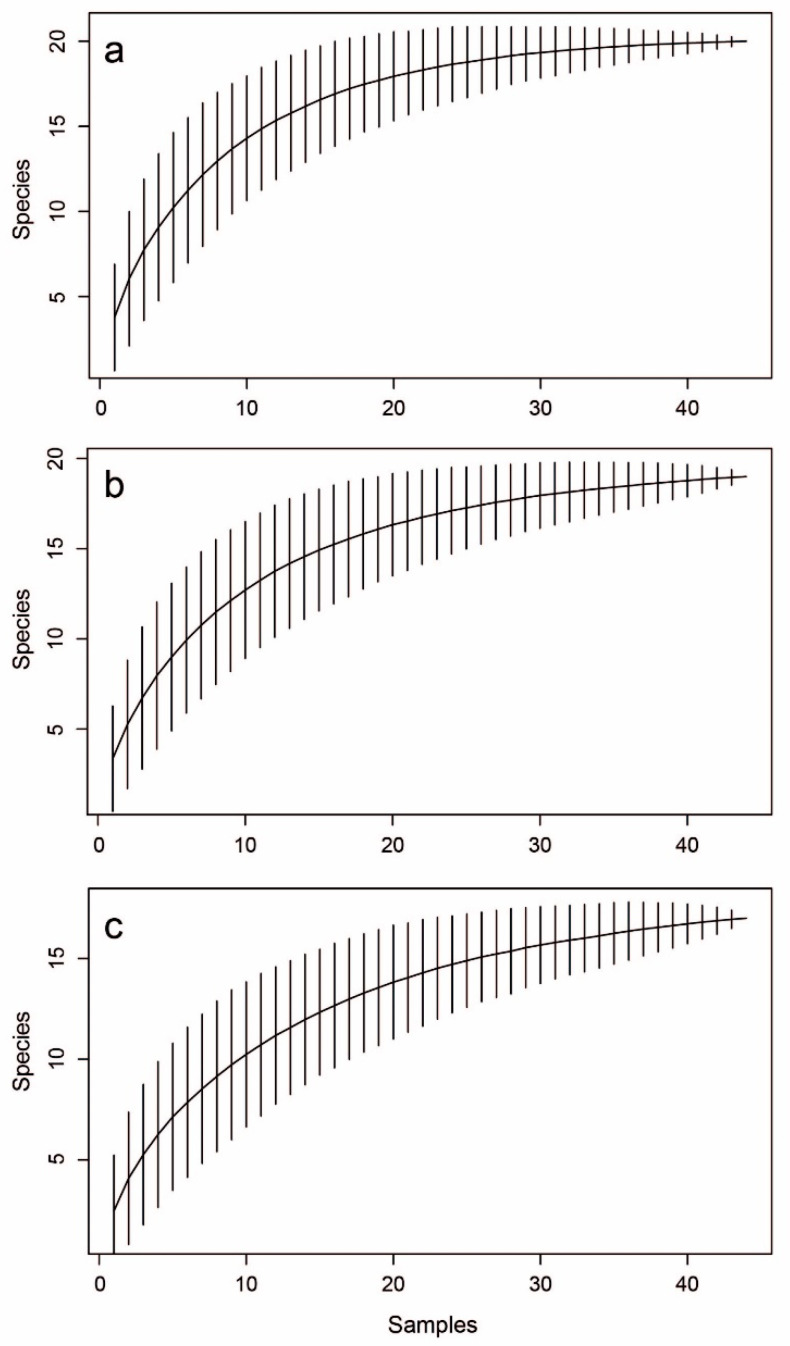
Species accumulation curves for (**a**) all samples; (**b**) specimens collected by hand net; (**c**) specimens collected from stones.

**Figure 3 insects-16-00723-f003:**
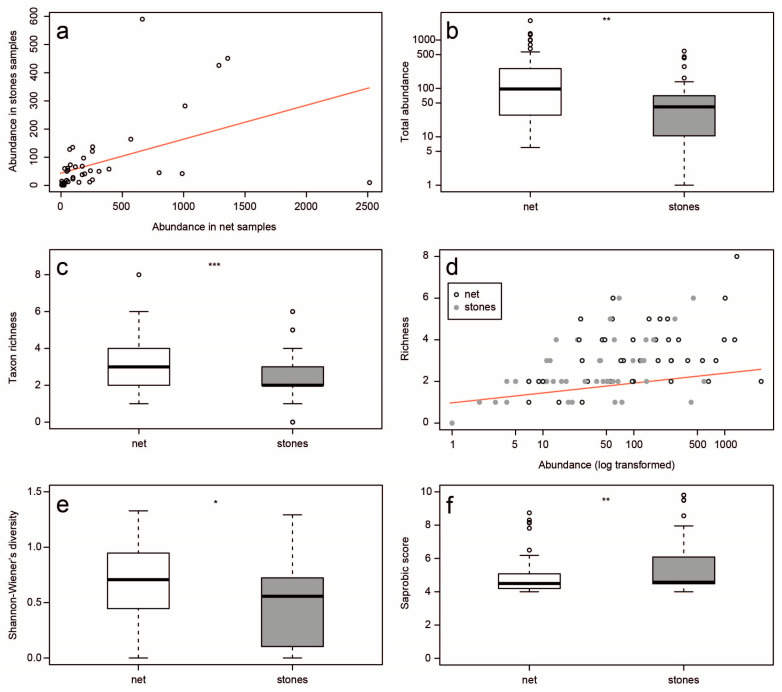
Comparison of the net vs. stone techniques. (**a**) Total abundance per sampling event; (**b**) total abundance of collected mayflies (paired); (**c**) taxon richness (paired); (**d**) taxon richness as a function of total abundance; (**e**) Shannon–Wiener’s diversity (*H*’) (paired); (**f**) saprobic score (paired). Significance levels: * *p* < 0.05, ** *p* < 0.01, *** *p* < 0.001.

**Figure 4 insects-16-00723-f004:**
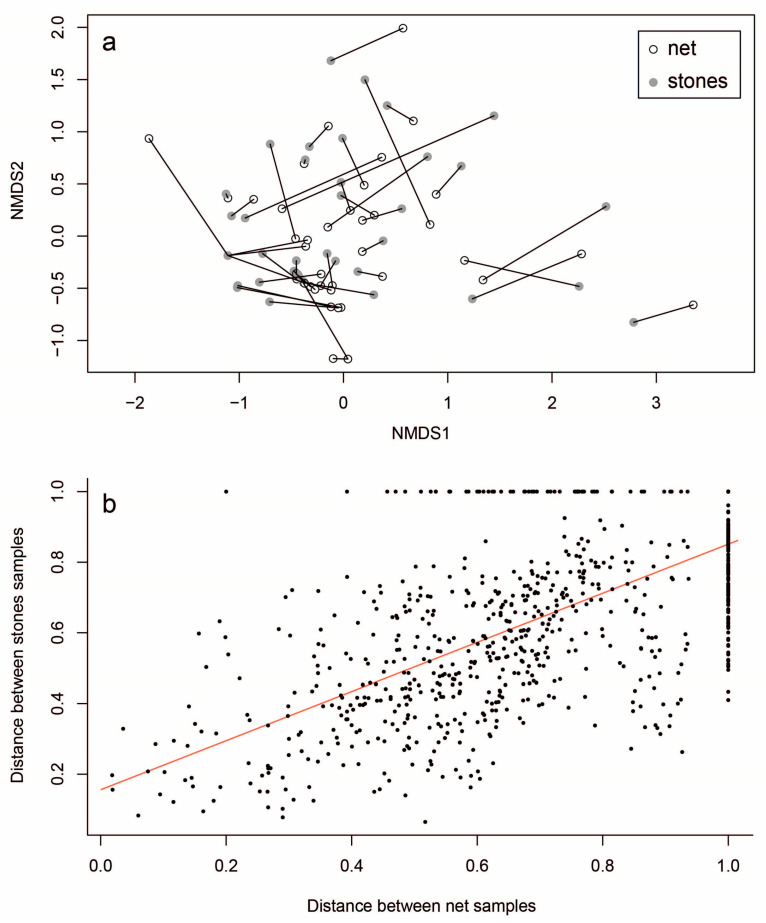
(**a**) NMDS ordination for community structure in net-collected (white) and stone-collected (grey) samples. Pairs are linked by line; (**b**) pairwise dissimilarity amongst all samples.

**Table 1 insects-16-00723-t001:** Occurrence (number of samples) of each taxon in each sampling technique and biological traits used in further analyses. Results of Wilcoxon test (*P*, V), presented for each species based on relative abundance data (total n = 44 sites). Selected parameters include microhabitat preference (sensu [12]) and locomotion type (sensu [3,26]). Microhabitat abbreviations: B = boulders; C = coarse gravel and cobbles; F = fine gravel; M = macrophytes; W = wood material. Locomotion abbreviations: B = burrowing; P = sprawling; W = swimming.

Species	Occurrence			*P*	V	Microhabitat Specialisation	Locomotion Type
	(Total)	(Net)	(Stones)				
BAETIDAE							
*Cloeon* sp.1	4	4	0			mainly M	W, some P
*Cloeon* sp.2	7	7	1	0.02	28	mainly M	W, some P
*Procloeon* sp.	2	2	0			diverse	W, some P
*Baetis golanensis*	14	12	11	0.58	22	M (?)	W, some P
*Baetis monnerati*	35	33	32	0.11	217	M (?)	W, some P
*Baetis aureus*	8	7	3	0.31	26	M (?)	W, some P
*Baetis pacis*	5	1	4			M (?)	W, some P
*Baetis samochai*	3	2	2			M (?)	W, some P
*Baetis (Rhodobaetis) noa*	6	6	2	0.03	21	M (?)	W, P
*Alainites gasithi*	5	5	0			mainly M	B, P
HEPTAGENIIDAE							
*Anapos kugleri*	12	11	8	0.62	32		
*Electrogena galileae*	5	4	5			C, little W	P
*Rhithrogena znojkoi*	5	4	4			C	P
*Ecdyonurus asiaeminoris*	1	0	1			C	P
*Epeorus zaitzevi*	3	1	3			B, C	P
PROSOPISTOMATIDAE							
*Prosopistoma oronti*	5	4	4			B, C	
CAENIDAE							
*Caenis* spp.	35	34	20	<0.01	576	diverse	P, other types
LEPTOPHLEBIIDAE							
*Choroterpes ortali*	5	5	4			C, F	mostly P
Choroterpes picteti	2	2	1			C, little B	mostly P
OLIGONEURIIDAE							
*Oligoneuriopsis orontensis*	2	2	2			diverse	

**Table 2 insects-16-00723-t002:** Merits of and considerations in the assessment of net vs. stone techniques for mayfly sampling. The table summarises the motivation for selecting either technique based on the study purpose, as well as highlighting limitations that should be acknowledged when interpreting results.

	Net Technique	Stone Technique
Community structure	Well reflected in the collected assemblage.	Well reflected in the collected assemblage.
Taxon richness	High.	Low.
Number of collected individuals	Large numbers—optimal for abundance-based analyses or for studies that require large samples of a population (e.g., for genetic study) or for rearing.	Smaller numbers.
Nymph locomotion type	Ideal for free-swimmers (e.g., Baetidae) and mud-dwellers (e.g., Caenidae).	Ideal for clingers (e.g., Heptageniidae).
Focus	Unnecessary overkill.	Only focal species.
State of collected specimens	Often damaged during transfer.	Specimens picked individually and kept in good physical state.
Required field skills	Experience in standard aquatic net sampling.	Dexterity necessary; individuals may drop from stones before they are handpicked. Taxa may be overlooked in the field.
Fieldwork	Little time spent on collecting and sorting in the field. Requires large amounts of ethanol.	Time-consuming fieldwork. Requires small amounts of ethanol.
Lab work	Time-consuming lab sorting.	No lab time spent on sorting and removing unwanted organisms and debris.
Resulting data	Individuals of lower quality for identification and for taxonomic studies.	Usually does not result in measurable and comparable data.

## Data Availability

The original contributions presented in this study are included in the article. Further inquiries can be directed to the corresponding author.

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
