# Peer review of "Sampling Techniques Affect Mayfly Nymph Community Indices and May Bias Bioassessments"

_insects, 2025, doi:10.3390/insects16070723_

Round 1
Reviewer 1 Report
Comments and Suggestions for Authors
The study titled “Sampling techniques affect mayfly-nymph assemblages and may bias bioassessments” was aimed at comparing two sampling techniques net sampling vs hand picking (stone) sampling. The study was well thought out and the manuscript well structured except for the materials and methods that was not sectionalised. I would suggest you section the materials and method sections, as lumping all sub-section such as study are/sites, sampling techniques, data analyses, etc makes the manuscript difficult to understand. I didn’t see any detailed description of study area and sites. What informed your selection of the 15 streams, and how many sites and biotopes/microhabitats did you sampled per stream? I think details on the merits and demerits of both sampling techniques under comparison will make more sense to any reader. Further, if you were set to compare sampling techniques why did you also look at the community structure and functional ecology of the mayflies? I think with this, you might need to recraft your manuscript title to reflect the content of the manuscript. Then, the introduction needs to be more direct, as most of your arguments were mainly on the sampling techniques without recourse to the community structure and functional ecology you also included in the body of the manuscript. Again, you need to diversify your literature search as most of your literature were too local, at least search for literature from up to four continents in which macroinvertebrates-based studies have been carried out to get a glimpse of what is happening all over for a good judgement. For the results section what informed your selection of traits attributes? You selected only two trait classes: microhabitat specialisation and locomotion type, why? Was your trait award done apriori or aposteriori? If any of them is yes, please explain. Again, the results speak less of the comparison of sampling techniques and more of the community structure and functional approach, why? I think you need to revisit this section or modify the title to reflect what you actually did. For the discussion, I didn’t look at it as I am not quite okay with the results to inform my judgement of the discussion of your findings.

Author Response
Comment 1: I would suggest you section the materials and method sections, as lumping all sub-section such as study are/sites, sampling techniques, data analyses, etc makes the manuscript difficult to understand.
Response 1: we followed the comment and resectionised the M&M section with subsection 2.1–2.5.
Comment 2: I didn’t see any detailed description of study area and sites. What informed your selection of the 15 streams, and how many sites and biotopes/microhabitats did you sampled per stream?
Response 2: we provided more information on the selected streams in section 2.1.
Comment 3: if you were set to compare sampling techniques why did you also look at the community structure and functional ecology of the mayflies? I think with this, you might need to recraft your manuscript title to reflect the content of the manuscript.
Response 3: Community structure and functional ecology are common tools to estimate invertebrate diversity, are they affected by the sampling technique. We sampled using two techniques, and then analised the results using various tools, including community structure and functional ecology, to demonstrate how these are affected by the sampling technique. We feel that the title is pretty informative and accurate, and the Reviewer gave no alternative suggestion, but we are open to consider and idea by the Reviewer or the Editor.
Comment 4: you need to diversify your literature search as most of your literature were too local, at least search for literature from up to four continents in which macroinvertebrates-based studies have been carried out to get a glimpse of what is happening all over for a good judgement.
Response 4: surprisingly, literature that provides accurate description of sampling techniques and their consequences is pretty rare. It usually relies on defined studies that were conducted in a given place (just like our study), such as US, Europe, or South Africa. We included all the relevant literature that we have found, but we are happy to extend the search if needed.
Comment 5: For the results section what informed your selection of traits attributes? You selected only two trait classes: microhabitat specialisation and locomotion type, why? Was your trait award done apriori or aposteriori? If any of them is yes, please explain.
Response 5: data regarding Israel’s local mayfly fauna is pretty incomplete, and the analysis of other characters was impossible due to lack of data. We used reasonable characters that were available for our set of organisms. All characters were determined apriori, based on literature and personal past data. We now explain this in the revised version (section 2.5).
Comment 6: the results speak less of the comparison of sampling techniques and more of the community structure and functional approach, why? I think you need to revisit this section or modify the title to reflect what you actually did. For the discussion, I didn’t look at it as I am not quite okay with the results to inform my judgement of the discussion of your findings.
Response 6: the Results section includes the community structure and all the indices that were calculated. We believe that the Results section is not the appropriate place to interpret the results, but the Discussion. In the Discussion section we present all the interpretation regarding the results’ meaning, in the light of the comparison between the two sampling techniques, exactly as the Reviewer asks.
Comments from the attached commented pdf file:
Comment 7: the Reviewer asked to rephrase this sentence for clarity: All these scores were compared in a paired design using Wilcoxon’s text as none of them significantly distributed.
Response 7: we rephrased it as follows: All these scores were compared in a paired design, i.e. the net score and the stone score were compared for each assemblage. As the distribution of the resulted scores was usually not normal, we used Wilcoxon’s test.
Comments 8 and 9: in the opening paragraph of the Results, the Reviewer asks to move one part to the M&M and another part to the Discussion.
Responses 8 and 9: we add a paragraph in section 2.5 to explain the method and the motivation to use it. In the Results section, we keep the report on the results themselves and associated Jackknife estimation of taxon richness. We first give the number of species (clearly this is a result), and only then can run analyses on these numbers and calculate the “true” number of species. We see no need to move parts of this paragraph to the Discussion (also in accordance to the comment of Reviewer 2 regarding this paragraph).
Comment 10: I thought you mentioned this in your previous sections? Please delete. (regarding two sentences in the legend of Table 1).
Response 10: we deleted these sentences.
Grammatical and typing errors were corrected based on the Reviewer suggestions.
Reviewer 2 Report
Comments and Suggestions for Authors
The manuscript titled "Sampling techniques affect mayfly-nymph assemblages and may bias bioassessments" by Zohar Yanai and Netta Dorchin investigates the impact of different sampling techniques on the assemblages of mayfly nymphs in stream habitats. The study compares two commonly used methods: aquatic sweep nets and manual collection from stones. The authors conducted a field survey across 15 streams in Israel, sampling each stream twice using both techniques. Results indicated that the two methods produced different values for various biological indices, such as total abundance, taxon richness, Shannon-Wiener’s diversity index, and saprobic index. The study concluded that reliance on a single collection method can bias ecological assessments. The authors emphasized the importance of either: selecting an appropriate sampling technique that fits with their research objectives or combining multiple methods to ensure taxonomic completeness.
Overall Impressions:
Overall, the manuscript is well-written and presents a thorough investigation into the effects of sampling techniques on mayfly nymph assemblages. The study is methodologically sound and quantifies biases introduced by different sampling methods; however, there are some organizational issues (such as methods being described in the Results section) which could be improved for better clarity and flow.
Furthermore, there are instances of imprecise language that need to be addressed to enhance the manuscript's readability and accuracy.
The novelty of the paper (which describes a possible source of bias in bioassessment) is limited, and the paper advances the field of bioassessment by only a small increment; however, the concise presentation of the study and low page count enhance its publishability.
I recommend revisions, followed by a second round of review to verify that referee’s comments are adequately addressed.
Specific Comments and Editorial Suggestions:
-Line 57, mayfly nymphs “exhibit variable functional diversity”: If we don’t consider exceptionally long timeframes that are relevant to processes like natural selection and genetic drift, is functional diversity not a constant? What is the benefit of considering functional diversity to be variable?
-Line 156, “scores were compared in a paired design using Wilcoxon’s text as none of them significantly distributed”: The Wilcoxon Signed Rank Test evaluates the hypothesis that the median difference between two paired samples is zero. It is a non-parametric test used when the distribution of differences between pairs is severely non-normally distributed, making it an alternative to the paired t-test. The statement that the index values were not significantly distributed is nonsensical.
Line 167/168, “The two matrices were then compared using Mantel Test, to identify any controversy between the two independent distance structure”: the Mantel Test is used to test the null hypothesis of no correlation between distance matrices. It cannot identify or evaluate controversy.
Line 176: Jackknife technique is introduced as a method of estimating total mayfly species diversity. This technique should be introduced in the methods section, and some elaboration of exactly how jackknifing was performed would enhance readability.
Lines 186 to 188, “The number of individuals yielded by both sampling techniques correlated well (Pearson’s correlation, r=0.71, n1,2=44, P<0.001; Figure 3a), suggesting that both reflected the true abundance, i.e. both were random subsets of the total community.”: If the two sampling techniques provided similar estimates of abundance, then we can conclude that they are correlated or have similar sampling biases, but correlation alone would not be sufficient to conclude that the techniques provide estimates of abundance that are true representations of mayfly abundances in the sampled habitats.
Figure 3: I think the 1:1 plots, as in 3a are the most straightforward for illustrating comparability and equivalence, because they directly show the residuals around the 1:1 line and how these residuals vary over the ranges of indices calculated in the study. Is there a reason why a 1:1 plot was used for abundance, but box-and-whisker plots are used for the other indices?
Figure 4: it struck me that Bray-Curtis dissimilarity was 1.0 for around 50 pairwise contrasts. What is the significance of this value? What does 1.0 actually mean? (I think it means that the two samples being compared have no common taxa). Is it worth specific mention in the Results section that the different techniques had no overlapping taxa in this many cases?
Lines 220-221, “The relative abundance of each taxon was compared between sampling methods within each sampling event”: belongs in Methods section, not Results
Lines 230-231, “Statistical analyses were only performed for species with ≥6 observations. Species traits were obtained from www.freshwaterecology.info”: belongs in Methods section, not Results.
Table 1: Please spell-out all the column headers, and please elaborate regarding how this table quantifies the “association of species with sampling techniques”
Lines 299/300: Epeorus is a clinger, crawler, and grazer, correct? I don’t think it is a “sprawler”….Sprawlers typically live on fine sediments. They “sprawl-out” extending appendages to avoid sinking into the fine materials. Epeorus, at least the ones I’m familiar with in Canada, do not have this habit.
317-319: I wonder if it would make sense to elaborate on the practical realities of trying to amalgamate data from the two different methods. If a researcher was to use both, for reasons of taxonomic completeness, especially concerning rare taxa, would combining methods and uniting the data cause any numerical/statistical issues?
Author Response
Comment 1: Line 57, mayfly nymphs “exhibit variable functional diversity”: If we don’t consider exceptionally long timeframes that are relevant to processes like natural selection and genetic drift, is functional diversity not a constant? What is the benefit of considering functional diversity to be variable?
Response 1: a better phrasing is “exhibit a variety of functional traits”, and the text changed accordingly. We refer to common practices in freshwater ecology, i.e. categorising species according to their ecological functions (predators, detritivores, grazers, crawlers, swimmers, etc.).
Comment 2: Line 156, “scores were compared in a paired design using Wilcoxon’s text as none of them significantly distributed”: The Wilcoxon Signed Rank Test evaluates the hypothesis that the median difference between two paired samples is zero. It is a non-parametric test used when the distribution of differences between pairs is severely non-normally distributed, making it an alternative to the paired t-test. The statement that the index values were not significantly distributed is nonsensical.
Response 2: the Reviewer is correct. We rephrased this sentence, and it now reads: As the distribution of the differences between the pairs of resulted scores was usually not normal, we used Wilcoxon’s test.
Comment 3: Line 167/168, “The two matrices were then compared using Mantel Test, to identify any controversy between the two independent distance structure”: the Mantel Test is used to test the null hypothesis of no correlation between distance matrices. It cannot identify or evaluate controversy.
Response 3: we rephrased the text to be more accurate: “We then used the Mantel Test to test for any correlation between the two independent distance structures.”
Comment 4: Line 176: Jackknife technique is introduced as a method of estimating total mayfly species diversity. This technique should be introduced in the methods section, and some elaboration of exactly how jackknifing was performed would enhance readability.
Response 4: we elaborated on the motivation and way we used Jackknife1 in a new paragraph in section 2.5.: “We sought to estimate the true number of mayfly taxa at each sampling event, i.e. to ensure that our sampling effort was sufficient in detecting most of the present may-fly taxa. For this purpose, we used the non-parametric diversity estimator Jackknife1. We estimated the expected number of taxa for each sampling method separately, in aim to indicate any potential gap in their efficiency in reflecting the “true” taxon rich-ness (regardless of species identity which was addressed separately, see below).”
Comment 5: Lines 186 to 188, “The number of individuals yielded by both sampling techniques correlated well (Pearson’s correlation, r=0.71, n1,2=44, P<0.001; Figure 3a), suggesting that both reflected the true abundance, i.e. both were random subsets of the total community.”: If the two sampling techniques provided similar estimates of abundance, then we can conclude that they are correlated or have similar sampling biases, but correlation alone would not be sufficient to conclude that the techniques provide estimates of abundance that are true representations of mayfly abundances in the sampled habitats.
Response 5: the Reviewer is correct. We used poor wording and failed to deliver the message. We changed the phrasing: “...both were consistent in sampling a subset of the total community.”
Comment 6: Figure 3: I think the 1:1 plots, as in 3a are the most straightforward for illustrating comparability and equivalence, because they directly show the residuals around the 1:1 line and how these residuals vary over the ranges of indices calculated in the study. Is there a reason why a 1:1 plot was used for abundance, but box-and-whisker plots are used for the other indices?
Response 6: we attempted to pick the figure type most appropriate to the analysis performed. For correlation, like in 3a, 1:1 plot represents the results pretty well. We think that in the case of pairwise comparison (Wilcoxon test) box-plots best represent the comparison between the two groups. It is possible to change the box plots to a 1:1 plots, if this better delivers the message, but we think that difference in median values and distribution of scores is better shown in a box plot.
Comment 7: Figure 4: it struck me that Bray-Curtis dissimilarity was 1.0 for around 50 pairwise contrasts. What is the significance of this value? What does 1.0 actually mean? (I think it means that the two samples being compared have no common taxa). Is it worth specific mention in the Results section that the different techniques had no overlapping taxa in this many cases?
Response 7: to be accurate, the meaning is that many stone samples had 0 similarity (were completely different), AND many net samples had 0 similarity. Notice that y axis is for stone difsimilarity and x axis for net dissimilarity. The cases of pairs that were completely different in both stone AND net samples cannot be told from the plot – these are the points placed in the very far end, where x=0 and y=0. This is in fact some portion of the paired samples, but not a big portion. In other words: some pairs had 0 similarity in stones, but not in net, and vice versa. This is probably a result of natural and ecological causes, that are reflected in the biological community. We hope that this is clearer now.
Comment 8: Lines 220-221, “The relative abundance of each taxon was compared between sampling methods within each sampling event”: belongs in Methods section, not Results
Response 8: we modified the sentence and it now reads: “ Analysing the relative abundance of each taxon by sampling method revealed...”. The M&M section already mentioned this method: “Since the net method yielded significantly more specimens than the stone technique, intra-specific abundance analyses were conducted on the relative abundance within each method.”
Comment 9: Lines 230-231, “Statistical analyses were only performed for species with ≥6 observations. Species traits were obtained from www.freshwaterecology.info”: belongs in Methods section, not Results.
Response 9: we moved this information to the Methods section.
Comment 10: Table 1: Please spell-out all the column headers, and please elaborate regarding how this table quantifies the “association of species with sampling techniques”
Response 10: we modified the name of the table, i.e. the first sentence of the legend, like so: “Occurrence of each taxon in any sampling technique (or both), alongside the biological traits that were used in further analyses.” Spelling-out the header results in too-complicated and long table structure. We tried playing with it, but we think that the legend is very clear and close enough to the headers. If MDPI’s proofing office can offer a readable and pleasant presentation of the table, we are happy to accept it in a spelled-out format.
Comment 11: Lines 299/300: Epeorus is a clinger, crawler, and grazer, correct? I don’t think it is a “sprawler”….Sprawlers typically live on fine sediments. They “sprawl-out” extending appendages to avoid sinking into the fine materials. Epeorus, at least the ones I’m familiar with in Canada, do not have this habit.
Response 11: we obtained trait information from freshwaterecolgy.info. We try to be consistent and objective in assigning the traits, and not providing new information in our manuscript since it was not the scope of this study. In freshwaterecolgy.info the only Epeorus species that has locomotion information is E. assimilis, and it is defined as sprawler.
Comment 12: lines 317-319: I wonder if it would make sense to elaborate on the practical realities of trying to amalgamate data from the two different methods. If a researcher was to use both, for reasons of taxonomic completeness, especially concerning rare taxa, would combining methods and uniting the data cause any numerical/statistical issues?
Response 12: it is a fair point. Unfortunately, our data and analyses do not give us enough information to point on such problems (or solutions to them). We think such aspect may be a good motive for future research.
Reviewer 3 Report
Comments and Suggestions for Authors
Manuscript Peer Review Report
Manuscript ID - insects-3602067
Insects (ISSN 2075-4450)
special issue: Aquatic Insects: Ecology, Diversity and Conservation
Overall comments:
This manuscript describes a comparison of two common aquatic invertebrate field collection approaches within the context of their applicability to different survey or assessment goals. The study design and statistical analyses of the data presented in this manuscript appear to be sound and the results of this study may inform future decisions when planning for studies to characterize aquatic habitats or ecological parameters of aquatic invertebrate communities. In general, the
manuscript is well written, however it is suggested that the authors carefully review verb agreement and verb tense use consistency throughout manuscript before resubmitting. Specific line by line comments are provided below for the authors to consider for improvement.
Line-by-line comments:
Line 75 (and elsewhere throughout the manuscript):
Avoid use of passive voice (e.g., line 75 “we have tested”) when describing the activities that took place as part of this study.
Line 194-195: Consider revising:
“Shannon–Wiener’s diversity index (H’) that was also calculated for all samples and compared between the two methods and showed…..”
Figure 3: “Stones” vs “Stone”
see comment provided for this same descriptor inconsistency issue encountered in Table 2.
Figure 3, panel d:
The similarity in shape and shading of the graph plot symbols makes it difficult for the reader to discern which dots represent “net” and which dots represent “stones” in the figure. Consider revising how these data are represented (e.g., black circles and white circles.
Figure 4, panel a:
Same comment as above. Instead of grey, consider solid black circles.
Lines 227 – 237, Table 1:
The column headings applied here (“n(total)”, “n(net)”, “n(stones”) are misleading. Even though this abbreviation is described in the table caption, it is easy to confuse “n” as representing numbers of individuals, since this abbreviation commonly represents that parameter. The characteristic I suspect that you are instead representing here is the observed number of sampling attempts where the species was encountered (in other words, species presence observations) across all sites samples and by sampling gear type.
Define “P” and V in the table caption.
Line 265-267: Table 2
Table caption – “Stones” vs “Stone”. Based on descriptor use elsewhere in the manuscript, suggest changing the title of this table to: “Merits of and considerations in net vs. stones technique.”
Row labeled “Attention”: perhaps there is a more appropriate descriptor for this characteristic, such as “Field Personnel Skill”. Are there any considerations associated with the Net Technique?
What, if any are the differences in the total time the field technician is required to spend in the field to collect enough data to approach the asymptote in Figure 2 (e.g., effectively characterize the stream community) for these two approaches?
Re: Suitability of resulting data for nets: suggest revising text to include what constitutes “low quality” (e.g., physical damage from collection handling may result in loss if key taxonomic features used for taxonomic identification)
Author Response
Comment 4: In general, the manuscript is well written, however it is suggested that the authors carefully review verb agreement and verb tense use consistency throughout manuscript before resubmitting.
Response 4: Comment implemented. We went over the entire manuscript and improved the text.
Comment 5: Line 75 (and elsewhere throughout the manuscript):
Avoid use of passive voice (e.g., line 75 “we have tested”) when describing the activities that took place as part of this study.
Response 5: We implemented this comment in most places. In places where we opted for the passive voice (such as the Material and Methods section) this is in accordance with recent articles in this journal, including Liu et al. 2025 and Popović et al. 2025, in the same special issue.
Comment 6: Line 194-195: Consider revising: “Shannon–Wiener’s diversity index (H’) that was also calculated for all samples and compared between the two methods and showed…..”
Response 6: Comment implemented (currently on lines 207–208).
Comment 7: Figure 3: “Stones” vs “Stone”. see comment provided for this same descriptor inconsistency issue encountered in Table 2.
Response 7: Implemented. “stones” (the plural form) is now used only in the context of the “stones technique”.
comment 8: Figure 3, panel d: The similarity in shape and shading of the graph plot symbols makes it difficult for the reader to discern which dots represent “net” and which dots represent “stones” in the figure. Consider revising how these data are represented (e.g., black circles and white circles.)
Response 8: Comment implemented. The two types of symbols in Figure 3 panel d are now clearly different.
Comment 9: Figure 4, panel a: Same comment as above. Instead of grey, consider solid black circles.
Response 9: Comment implemented.
Comment 10: Lines 227 – 237, Table 1: The column headings applied here (“n(total)”, “n(net)”, “n(stones”) are misleading. Even though this abbreviation is described in the table caption, it is easy to confuse “n” as representing numbers of individuals, since this abbreviation commonly represents that parameter. The characteristic I suspect that you are instead representing here is the observed number of sampling attempts where the species was encountered (in other words, species presence observations) across all sites samples and by sampling gear type.
Response 10: Comment implemented. The title and legend of the table were changed as suggested.
Comment 11: Define “P” and V in the table caption.
Response 11: P and V are the significance level and statistic of the Wilcoxon test. This is now indicated in table caption.
Comment 12: Line 265-267: Table 2. Table caption – “Stones” vs “Stone”. Based on descriptor use elsewhere in the manuscript, suggest changing the title of this table to: “Merits of and considerations in net vs. stones technique.”
Response 12: Comment implemented.
Comment 13: Row labeled “Attention”: perhaps there is a more appropriate descriptor for this characteristic, such as “Field Personnel Skill”. Are there any considerations associated with the Net Technique?
Response 13: we changed “attention” to “required field skills” and modified the text for the net technique.
Comment 14: What, if any are the differences in the total time the field technician is required to spend in the field to collect enough data to approach the asymptote in Figure 2 (e.g., effectively characterize the stream community) for these two approaches?
Response 14: We added some text to the table to clarify this point.
Comment 15: Re: Suitability of resulting data for nets: suggest revising text to include what constitutes “low quality” (e.g., physical damage from collection handling may result in loss if key taxonomic features used for taxonomic identification)
Response 15: We modified the text in the table and added details on lines 332–335.
Round 2
Reviewer 1 Report
Comments and Suggestions for Authors
Response from authors: Response 6: the Results section includes the community structure and all the indices that were calculated. We believe that the Results section is not the appropriate place to interpret the results, but the Discussion. In the Discussion section we present all the interpretation regarding the results’ meaning, in the light of the comparison between the two sampling techniques, exactly as the Reviewer asks.
My Comment: If the Results section is not the appropriate place to interpret the results, where should the findings of a research be interpreted? Data analysed must be interpreted in the results while Discussion of the implications of the results are done in the Discussion section. Please revisit my comments and address accordingly.
Then, as I stated earlier the title must be modified to reflect the content of the manuscript. I am wondering why authors are expecting reviewers to suggest titles for a research done by them. Please rephrase your title accordingly to reflect what you have done.
Please also tell us why you use apriori instead of a posteriori.
Author Response
Comment 1: If the Results section is not the appropriate place to interpret the results, where should the findings of a research be interpreted? Data analysed must be interpreted in the results while Discussion of the implications of the results are done in the Discussion section. Please revisit my comments and address accordingly.
Response 1: We disagree with the reviewer on this point. We contend that, as is customary in research articles, the results section should be dedicated to reporting the results whereas the place for their interpretation is the discussion. We do not understand why we are asked to deviate from this pretty much universal scheme of a scientific manuscript.
Comment 2: Then, as I stated earlier the title must be modified to reflect the content of the manuscript. I am wondering why authors are expecting reviewers to suggest titles for a research done by them. Please rephrase your title accordingly to reflect what you have done.
Response 2: We do not understand why the reviewer finds the title problematic. We believe it reflects the content and essence of this study accurately as we already explained in our response in the first round of review. This study demonstrates that scores of mayfly community structure and functional ecology are affected by the sampling technique employed, so why not say that explicitly in the title? Moreover, in his/her original comment the reviewer wrote “I think you might need to recraft your manuscript title…”, which implies a suggestion rather than a demand, and is a matter of opinion or style. We think the title is accurate and to-the-point, and therefore ask to leave it as is.
Comment 3: Please also tell us why you use apriori instead of a posteriori.
Response 3: All characters were determined apriori, based on the literature and on personal information derived from past experience. We added some text in section 2.5 to explain the choice of the particular traits based on their reliability and explain the constraints in using additional traits, many of which are unavailable for the local taxa, which precludes a meaningful analysis.
Round 3
Reviewer 1 Report
Comments and Suggestions for Authors
Dear authors, thanks for submitting the revised version of your manuscript. Though, you have been able to improved the manuscript in the current version. However, most of my comments were not addressed, what I can see is more of sentence structure revision, the main science which I raised concern about earlier have not been addressed. For instance, in my comment 2, I quote "Comment 2: I didn’t see any detailed description of study area and sites. What informed your selection of the 15 streams, and how many sites and biotopes/microhabitats did you sampled per stream?" and your response was "Response 2: we provided more information on the selected streams in section 2.1". I didn't seen any information you added here, please check and rework, please.
I also raised concern about the title not aligning with the message passed in the body of the work. Your title was more or less speaking to sampling technique of mayfly assemblage, I am wondering how this translate to analysing community and functional structures of the said insect order. Please coin your title around the exploration of the assemblage and functional structures of the insect you studied, please. I think this should help, otherwise, suggesting titles for you is not my responsibility in my opinion. So, kindly recoin the title in your next revision for re-assessment, please.
Then, for the literature, you did not address that, please add literature from other parts of the world for global representation of your search.
Finally, for the results interpretation I raised concern about, your response was not satisfactory. What I mean by results interpretation is outlining the outcome of your statistical analysis, not discussion. Results are not interpreted in Discussion section as you claimed, rather results are been discussed in detailed in the Discussion section, by way of aligning your findings with other studies and stating the implications of such findings, please rework for re-assessment.
Comments on the Quality of English LanguageFew grammatical corrections will help.
Author Response
Comment 1: I didn’t see any detailed description of study area and sites. What informed your selection of the 15 streams, and how many sites and biotopes/microhabitats did you sampled per stream?
Response 1: In section 2.1 we describe the diversity of streams that were sampled in the study with respect of climate, hydrology (lowland vs. mountainous), and human disturbance. We added a complete list of the stream names, although we believe it will mean nothing to readers outside Israel.
Our selection was informed by our previous experience with streams in the country, that accommodate large mayfly communities and represent ecological diversity. This was also informed by previous findings (see line 106). As explained in the text, each stream was sampled in two sites (lines 108–109). The availability of microhabitats varied across sites but we made an effort to sample all of them (lines 109–111, 117–120).
Comment 2: I also raised concern about the title not aligning with the message passed in the body of the work. Your title was more or less speaking to sampling technique of mayfly assemblage, I am wondering how this translate to analysing community and functional structures of the said insect order. Please coin your title around the exploration of the assemblage and functional structures of the insect you studied, please. I think this should help, otherwise, suggesting titles for you is not my responsibility in my opinion. So, kindly recoin the title in your next revision for re-assessment, please.
Response 2: We argue that community structure and functional diversity are aspects of the sampled assemblages. An assemblage may be analysed in many ways, including total abundance and taxon richness. If our title included the phrase “community and functional structures”, we would necessarily ignore other aspects that we analysed. Nevertheless, we rephrases the title to read “Sampling techniques affect mayfly-nymph community indices and may bias bioassessments”.
Comment 3: Then, for the literature, you did not address that, please add literature from other parts of the world for global representation of your search.
Response 3: This comment is unclear to us as we refer to literature that spans across Australia, Canada, Chile, Greece, New Zealand, South Africa, Spain, Switzerland, the UK, and the US; as well as continent-wide studies from North America, Africa, and Europe. Naturally, our literature review included studies of sampling methodologies and protocols, and focused on regions that provided such data.
Comment 4: Finally, for the results interpretation I raised concern about, your response was not satisfactory. What I mean by results interpretation is outlining the outcome of your statistical analysis, not discussion. Results are not interpreted in Discussion section as you claimed, rather results are been discussed in detailed in the Discussion section, by way of aligning your findings with other studies and stating the implications of such findings, please rework for re-assessment.
Response 4: We maintain that we do provide interpretation of the results in the Results section. For example: “...suggesting that all species that occurred in the study sites were indeed sampled...” (lines 196–198); “...suggesting that both were consistent in sampling a subset of the total community alt-hough the non-selective net method yielded significantly more individuals...” (lines 206–208); “...such that the more individuals collected, the higher taxon richness was expected...” (lines 211–212). To address the reviewer’s comment we did add some text to emphasize the meaning of the results in some cases (lines 225; 229–231; 248–249).
Round 4
Reviewer 1 Report
Comments and Suggestions for Authors
Thanks, all my concerns have been addressed accordingly.
All the best.